# Evolving Role of RING1 and YY1 Binding Protein in the Regulation of Germ-Cell-Specific Transcription

**DOI:** 10.3390/genes10110941

**Published:** 2019-11-19

**Authors:** Izabella Bajusz, Surya Henry, Enikő Sutus, Gergő Kovács, Melinda K. Pirity

**Affiliations:** 1Biological Research Centre, Temesvári krt. 62, H-6726 Szeged, Hungary; henry.surya@brc.hu (S.H.); sutus.eniko@brc.hu (E.S.); kovacs.gergo@brc.hu (G.K.); 2Doctoral School of Biology, Faculty of Science and Informatics University of Szeged, Dugonics tér 13, H-6720 Szeged, Hungary

**Keywords:** germ cell differentiation, transcriptional regulation, polycomb repression, Rybp, ncPRC1, Nanog, Oct4, Sall4, ubiquitylation, retinoic acid, apoptosis, meiosis, transcriptome

## Abstract

Separation of germline cells from somatic lineages is one of the earliest decisions of embryogenesis. Genes expressed in germline cells include apoptotic and meiotic factors, which are not transcribed in the soma normally, but a number of testis-specific genes are active in numerous cancer types. During germ cell development, germ-cell-specific genes can be regulated by specific transcription factors, retinoic acid signaling and multimeric protein complexes. Non-canonical polycomb repressive complexes, like ncPRC1.6, play a critical role in the regulation of the activity of germ-cell-specific genes. RING1 and YY1 binding protein (RYBP) is one of the core members of the ncPRC1.6. Surprisingly, the role of Rybp in germ cell differentiation has not been defined yet. This review is focusing on the possible role of Rybp in this process. By analyzing whole-genome transcriptome alterations of the *Rybp*^-/-^ embryonic stem (ES) cells and correlating this data with experimentally identified binding sites of ncPRC1.6 subunits and retinoic acid receptors in ES cells, we propose a model how germ-cell-specific transcription can be governed by an RYBP centered regulatory network, underlining the possible role of RYBP in germ cell differentiation and tumorigenesis.

## 1. Introduction: Germline, Soma and Embryonic Stem Cells

Differentiation of germ cells has crucial importance in the survival of species. Production of healthy germ cells is the basis of fertility. Our life starts with the fusion of the female and the male germ cells, which together form the totipotent zygote. The myriads of cells of our body with different sizes, morphologies, and functions develops from this unique cell, which not only contains all genetic information, but also has the potential to use it all. This is why germ cells are generally considered to be totipotent, “the stem cells of the species” [1,2]. Germinal linage is often considered as unipotent, since the only possible fate of the male germinal lineage is to produce sperm, if controlled by the testicular microenvironment. However, embryonic primordial germ cells (PGCs) and spermatogonial stem cells (SSCs) (Figure 1) are able to change their fate and convert to pluripotent spontaneously in culture or during disease conditions like testicular carcinoma [3]. Germ cells express a series of pluripotency marker genes (e.g., *Pou5f1/Oct4, Klf4, Nanog*) and share certain characteristics of stem cells. As both pluripotency and germ cell marker genes are often upregulated in tumors [4] it is not surprising that germ cell fate determination attracts great attention.

Gametogenesis has been thoroughly studied in model organisms, and in some aspects, in vitro differentiation systems using embryonic stem (ES) cells [5,6] and induced pluripotent stem (iPS) cells [7,8]. By now, a lot of information has been gathered about the epigenetic regulation, the specific transcription factors, important signalization events, and key effectors necessary for commitment of stem cells to germline fate [9,10,11,12,13,14] from different model organism and by in vitro differentiation of stem cells to primordial germ-cell-like cells (PGCLCs) [9,10,11,12,13,14,15,16,17].

Germ cells profoundly differ from somatic cells. They set apart during early embryogenesis and from then on, this separation is irreversible. Undifferentiated germ cells, called PGCs, are considered to be unipotent, they proceed on their own developmental pathways and normally give rise to only more matured germ cells [18], mainly by consecutive mitotic divisions. In order to keep the number of chromosomes fixed when gametes of the two sexes fuse upon fertilization, PGCs must enter meiosis at certain points of their development, before completing gametogenesis. Meiosis is characteristic only for germ cells, and it never occurs in any somatic lineages. During their development, germ cells go through a complex epigenetic reprogramming process [19], while at the same time, imprinting of their genome might enable the epigenetic inheritance of certain environmental conditions [20,21,22].

Cells of the soma follow a completely different developmental path. As somatic cells differentiate from stem cells, during embryonic development, these cells lose pluripotency, while their differentiation reaches terminal stages. The germ cell differentiation program is closed for all somatic cells, repression of germ-cell-specific and meiotic genes is continuously maintained in the somatic lineage. Terminally differentiated somatic cells adopt their distinctive morphology and function, and this state is normally irreversible. Pluripotency factors are not expressed in somatic cells and terminally differentiated somatic cells often enter G_0_ phase and stop dividing. Proliferating somatic cells divide only by mitosis. Their chromosome number remains the same, and they always produce identical daughter cells due to the correct maintenance of their distinctive transcriptional pattern. Many tissues have high regenerative capacity and they can be renewed from adult stem cell reservoirs. Furthermore, many cell types, like red blood cells, have a very limited lifetime, sometimes only a couple of weeks, so they need to be constantly renewed. Germ cells differ profoundly from somatic cells, they are considered theoretically immortal, because they serve as precursors of the gametes. They are so precious that they have to be defended from environmental stress and possible DNA damages too, as they are the only cells which can be passed to the next generation by fertile individuals [23].

## 2. Differentiation of Germ Cells

There are two basic ways of the germ cell differentiation process: preformation and epigenesis [24,25,26,27]. In the case of preformation, which is the “inherited”, or deterministic way of germ cell determination, microscopically detectable, asymmetrically localized RNA-protein granules are found in the fertilized egg. A subset of blastomeres inherit these particles, often called the “germ plasm”, and these cells are the only ones, which can develop into germ cells. Germ granules have no surrounding membrane and contain mitochondria, RNAs and proteins necessary for germ cell development, and required for certain posttranscriptional regulatory processes specific to germ cells. In organisms of inherited germ cell determination, germ granules are present continuously throughout germ cell development except for the mature sperm [28]. Many famous model organisms have this type of germ cell determination, like the worm *Caenorhabditis elegans*, the fruit fly *Drosophila melanogaster* and the frog *Xenopus laevis* [24,29]. As preformation was the first type of germ cell determination studied, the hypothesis of the Weismann barrier was formed, based on these classical model systems [30]. According to this hypothesis, the germline is continuous, although all somatic cells originate from the germline, somatic cells never become germline cells again. 

The other type of germ cell determination, called epigenesis, is found in most mammals. It is described in detail in the course of mouse embryogenesis, when a group of somatic cells induce their neighboring cells of the blastula to become PGCs after implantation. In this case, cells formally belonging to the somatic lineage are induced to become germ cell precursors again, crossing the Weisman barrier. Although early development of a mouse embryo absolutely requires maternally provided determinants [31], no clear evidence is found that the localization of these determinants is important for the early germ cell determination process. It seems that the first embryonic differentiation steps and axis specification in mice occur in the absence of localized maternal determinants. The presence of maternal proteins and RNAs are detectable in the fertilized mouse zygote too, and can persist until the blastocyst stage, while zygotic gene expression starts, right after the first cell division [32,33].

Although not all organisms require germ granules to specify germ cell fate, all organisms depend on germ granules for germ cell function. Germ granules in mammals are only found at later developmental stages [28]. In mice, the first three cell divisions result in the 8-cell stage morula in which all cells are still equal, uniform and totipotent until the compaction phase. Compaction increases cell contacts, promotes adhesion and establishes apical-basal polarity in late 8-cell stage. This is the time when the first asymmetrically localized protein products appear in the mouse embryo and the cells become unequal [34]. Two subsequent cell divisions of these polarized cells result in an early blastocyst and by embryonic day 3.5 (E3.5) two distinctive cell types are formed. The thropho-ectoderm (TE) cells, which appear as a surrounding cell layer of the blastocyst, and inside of it, the cells of the inner cell mass (ICM). At the preimplantation stage, ICM is the classical source of ES cells, which can be cultured in vitro and expanded unlimitedly [5,6,35]. Later, primitive endoderm starts to appear beneath ICM. The germline, consisting only of a few cells at the beginning, is set apart from the somatic lineage only at implantation in the proximal epiblast. As in mammals germ cells are formed by an inductive mechanism, PGC differentiation largely depends on signals from the extraembryonic ectoderm [14]. Amongst somatic cells, a small group of alkaline phosphatase (AP) positive germ cell precursors appear at the most posterior side of the embryo at E6.25. These cells are called PGCs, the first identified founders of the gametes (Figure 1). These cells migrate towards the hindgut, while their number increases rapidly by mitotic cell divisions, by E8.5 they are embedded into the hindgut epithelium [36,37]. As gastrulation proceeds, PGCs keep migrating anteriorly to colonize the somatic compartment of the presumptive gonad. At this stage (E9–9.5) small germ granules are detectable in PGCs [38,39]. By E13 PGCs reach the genital ridge and become gonocytes (Figure 1). By that time, their inherited imprints are erased, and cells commit either a female or a male germ cell fate. In response to somatic signals, the germ cells enter meiosis E13.5 in females, while male germ cells go to mitotic arrest at E14.5, which is maintained until birth (Figure 1). Several rounds of mitotic cell divisions increase the number of SSCs in the testis until puberty when the meiotic waves of sperm differentiation are initiated [40,41]. During meiosis, spermatocytes assemble a specific type of germ granule called the chromatoid body [42]. Germ granule proteins play fundamental roles in regulating RNA processing. RNAs can be stored in the granules for weeks without being translated or being degraded. Specific control of gene expression at the RNA level is important in germ cells, and may be essential to preserve the plasticity of the germline genome [2]. VASA, PIWI, and NANOS homologs have been proposed to form a “conserved multipotency program” used not only by germ cells, but also by multipotent progenitors that are able to generat both somatic and germline tissues [43]. Not surprisingly, special surveillance mechanisms are also present in germ cells to ensure the integrity of their genome and defend them against transposons [44]. Some germ granule components like DDX4, the mouse homolog of VASA, and MAEL, Maelstrom spermatogenic transposon silencer, are important for efficient transposon silencing in mouse spermatocytes [45,46], apart from these specific DNA repair mechanisms that exist in PGCs [47], which require the reparation of double-strand breaks.

## 3. Germ Cell Differentiation Factors 

### 3.1. Maternally Contributed Factors

Many essential factors of germ cell differentiation have been identified during the years with profoundly diverse functions. The first class of germ cell determinants ever described is the group of maternally contributed factors, which are produced during oogenesis and deposited into the oocyte [48]. The RNAs and proteins of these factors are often specifically localized in the unfertilized egg forming morphogen gradients. Examples of this class include VASA, NANOS, and PIWI. These factors are often used as germ cell markers since the coding RNAs of these factors are exclusively found in germ cells and these conserved products are important in many species examined to set up the germ cell pluripotency program [43]. Many of the classical germ cell markers code for RNA binding proteins and germplasm constituents [49].

### 3.2. Diffusible Signaling Molecules

In mammals, diffusible signaling molecules are also necessary for the induction of germ cell fate. Bone morphogenic protein 4 (BMP4) is essential for the initiation of germ cell fate in mice [50]. BMP4 signaling from the extraembryonic ectoderm at E6.0 induces the neighboring epiblast cells to convert to germ cell fate. As a result of BMP4, signaling first high expression of interferon-induced transmembrane protein 3 (*Ifitm3*/*Fragilis)* is detectable, marking the onset of germ cell competence, later *Stella* expression is induced, which is a characteristic marker of germ cells [51]. PGCs are fully established as a small cluster of alkaline-phosphatase (AP) positive cells by E7.25 in the extraembryonic mesoderm [36,51]. Besides BMP4, retinoic acid (RA) signalization is also a critical component of germ cell fate determination. Exogenous administration of RA alone is also able to enhance the meiotic entry of germ cells [52].

### 3.3. Germ-Cell-Specific Transcription Factors

Besides to germplasm constituents and signaling molecules, specific transcription factors govern germ cell fate, like PR domain containing 1 with ZNF (Zinc finger) domain 1 (PRDM1, also known as BLIMP1) and PRDM14. These factors are zygotically expressed, and able to coordinately regulate the transcription of germ-cell-specific genes [12]. *Prdm1* is not only required for PGC specification, but its presence is persistently required during germ cell development [53]. Nanog homeobox (NANOG) is another important transcription factor expressed in PGCs in mice, although its role is not exclusively germ-cell-specific. NANOG was originally identified as an integral part of the core pluripotency network [7,54,55]. *Nanog* mutant cells are more prone to differentiate but do not become committed to any specific fate. ES cells are able to self-renew even if *Nanog* is transiently downregulated. *Nanog* expression fluctuates in stem cells [56]. The natural fluctuation of *Nanog* expression seems to be important, keeping the gate open towards differentiation in stem cells. A subpopulation of ES cells, which are characterized by low *Nanog* expression can differentiate more easily to different somatic lineages [57]. *Nanog* seems dispensable for maintaining the pluripotency of ES cells, which can self-renew in the permanent absence of *Nanog*, but it is essential for the establishment of ES cells from ICM [58,59] and required for the maturation of germ cells [56]. *Nanog* is expressed in normal fetal gonocytes during testis development in humans and often found to be active both in testicular carcinoma cells and germ cell tumors [60]. In an in vitro model system, it was found that the sustained presence of NANOG is important for germ cell fate initiation. Ectopic induction of *Nanog* expression alone can trigger the differentiation of PGCLCs from ES cells and the activation of *Prdm1* and *Prdm14* transcription independently of the activation of BMP4 and RA signaling [61].

### 3.4. Polycomb Repressive Complexes

Recently another group of factors was discovered which also plays important role in establishing and maintaining germ-cell-specific transcriptional activity. These proteins are members of different polycomb repressive complexes (PRCs). The genes coding for polycomb-type repressor proteins are highly conserved, but the PRCs formed in higher organisms are surprisingly diverse [62]. Polycomb proteins were discovered over 70 years ago and have been extensively studied since. The definition of the Polycomb group (PcG) was originally based on the regulation of homeotic genes in *Drosophila melanogaster*. These proteins are acting on the level of compacting chromatin structure [63] and holding the poised RNA polymerase at transcriptional start sites (TSSs), inhibiting its release [64,65,66]. PcG function is highly conserved. Homolog proteins were found in all examined species. PcG members were originally defined as epigenetic silencers. Their classical role is to secure maintained repression of key developmental genes, like the homeotic clusters in mammals, acting antagonistically with activators belonging to the trithorax group [67,68,69]. Nowadays, their role in many other aspects of the maintenance of cellular identity has been recognized. Mutations of PcG member genes disturb the differentiation of germ cells and the maintenance of germ-cell-specific gene expression [69,70], alter reprogramming processes [71], and affect the pathogenesis of different cancer types [72]. 

PRCs often interact with each other, and all of them are capable of covalent modification of histone tails. Enhancer of Zeste 2 (EZH2) containing, PRC2 complexes can specifically tri-methylate histone 3 (H3) at the 27th lysine (K27) [73,74,75], this methylation mark is recognized by the POLYCOMB/CHROMOBOX subunits of the canonical PRC1 complexes [76]. The catalytic RING subunits of the PRC1 complexes, in turn, can specifically ubiquitilate histone 2A (H2A) lysine 119 (K119) upon binding [77,78,79]. According to the hierarchical recruitment model of epigenetic repressor complexes, first sequence specific DNA binding proteins like PHO/YY1 (Pleiohomeotic; Yin and yang protein 1) recruit the EZH2 containing complexes, then H3 histones are specifically tri-methylated by PRC2 in the neighboring chromatin [80]. The H3K27 tri-methylation tag made by PRC2 can be recognized by the chromobox domain of PRC1 complexes and facilitates their binding [81] so the PRC1 complexes land last and compact chromatin.

Many different PRC1-type complexes were purified during the years [82], and it turned out that there is a specific subset of complexes, which does not have chromobox domain containing subunit at all, but contains RING1 and YY1 binding protein (RYBP) or YY1 associated factor 2 (YAF2) subunits instead [83,84]. These complexes are not able to recognize tri-methylated histones, but can be targeted differently based on either recognition of non-methylated CpG islands by Lysine (K)-specific demethylase 2B (KDM2B) [85,86] or specific DNA sequences by YY1 [87,88,89], transcription factor E2F6 alone [90] or together with MYC associated factor X (MAX) and MAX gene-associated protein (MGA) [91] or with Lethal (3) malignant brain tumor-like protein 2 (L3MBTL2) as well [92]. By now, the hierarchical model of PRC mediated repression turned upside down the classical viewpoint of how PRCs mediate repression of target genes. The RYBP containing non-canonical PRC1 (ncPRC1) complexes seem to be targeted independently of the presence of K27 tri-methylated histones and activity of PRC2s [93]. There is evidence for the opposite, H2A ubiquitylation helps PRC2 targeting [86,94]. According to the newest findings, PRC1 and PRC2 complexes can interact with each other independently from the histone modifications present, at least in certain cases [94,95].

The ncPRC1 complexes act as unusual repressors, because they often play a role in gene activation processes as well [66,96,97,98]. These complexes are subdivided, based on the Polycomb group ring finger (PCGF) subunits they contain. Different PCGFs all bind to the central RING1-RYBP catalytic core in a mutually exclusive way [83] and often have tissue-specific and molecular functions [99] (reviewed in [100]). Six PCGF paralog proteins were purified in the different ncPRC1 complexes, hence, there are six different ncPRC1 subtypes described and numbered after the PCGF subunit they contain. According to that, ncPRC1.6 is the specific PCGF6 subunit containing PRC1 type complex [83]. The subunit composition of different ncPRC1 complexes largely depends on the PCGF paralog subunit they contain. Different PCGFs selectively bind to certain protein partners [101] with their RING finger and WD40-associated ubiquitin-like (RAWUL) domain [102], which is sometimes called PCGF Ub-like fold discriminator (PUFD) module [101]. *Pcgf* paralogs show tissue and development-stage-specific expression pattern, which can be regulated by environmental stimuli [103]. The direct regulatory targets of different PCGFs also differ profoundly [99]. Knowing that, it is not surprising that knockout phenotypes of the mutations induced in the genes coding for subunits of the alternative PRC1-type complexes also highly vary (reviewed in [100]). Alternative ncPRC1 complexes are characterized with high target specificity and have few compensatory functions. ncPRC1.1 and ncPRC1.2 can compensate each other to certain extent in ES cells, but transposons are mainly repressed by ncPRC1.2 [99], while ncPRC1.6 is specifically involved in the regulation of meiotic and germ cell related genes [104] (Figure 2A).

#### Non-canonical PRC1.6, A Major Repressor of Germ Cell Fate

The RYBP containing ncPRC1.6 is one of the best-characterized germ-cell-specific epigenetic regulatory complex. Five of its subunits have been already connected to germ-cell-specific gene regulation or meiosis induction by independent findings. PCGF6 itself, the definitive subunit of the ncPRC1.6 complex, seems to be a key player organizing the repression of germ-cell-specific genes. PCGF6 is important to warrant the integrity of ncPRC1.6 [107], but *Pcgf6*^-/-^ mice are viable and fertile, although not born at the normal Mendelian ratio. Lethality amongst *Pcgf6^-/-^* embryos can be observed as early as 3.5 days post coitum, at the early blastocyst stage [104]. *Pcgf6* is highly expressed in ES cells. As demonstrated in multiple knockdown studies, *Pcgf6* gene is essential for the maintenance of ES cell pluripotency and essential to support effective self-renewal [107,108,109] and it is a key factor for iPS reprogramming [105]. Both the self-renewal capacity and the differentiation ability are seriously compromised in *Pcgf6^-/-^* ES cells [108,109]. *Pcgf6* is expressed predominantly in meiotic and post-meiotic male germ cells during development and knockdown of *Pcgf6* alters male germ cell differentiation in vitro in an immortalized spermatogenic cell line GC-2spd [110]. PCGF6 plays a direct role in regulating germ-cell-specific gene expression. Disruption of *Pcgf6* leads to the robust derepression of germ-cell-specific and meiotic genes in ES cells. E2F6, which is one of the four DNA binding subunits of ncPRC1.6 (Figure 2) was also identified as a repressor of a palette of germ-cell-specific genes [111]. Ten years later, it turned out that another specific DNA binding subunit of the complex, the MAX (Figure 2) protein, also acts as a repressor of germ-cell-specific genes in mouse ES cells [112]. The loss of *Max* function caused meiotic entry and consecutive cell death in mouse ES cells and germline stem cells [113,114,115]. According to the most recent data, the genomic binding of the ncPRC1.6 complexes is determined by not only E2F6 and MAX, but MGA. DNA Specific binding of the MAX-MGA module seems to be responsible for targeting PCGF6 to the promoters of germ-cell-specific and meiotic genes in close proximity of the TSSs [104]. L3MBTL2 protein is also required for targeting ncPRC1.6 [92]. L3MBTL2 is highly expressed in the testis, especially in pachytene spermatocytes. Mutation in *L3mbtl2* led to progressively decreased sperm counts, abnormal spermatozoa and increased γH2AX deposition in leptotene spermatocytes, defective crossing over and increased germ cell apoptosis in mice. Surprisingly the transcriptional activity of spermatogenesis associated genes has not changed significantly in the *L3mbtl2* mutant background [116].

The two alternate catalytic subunits of all PRC1 complexes, RING1 (Ring finger protein 1, also known as RING1A) and RNF2 (Ring finger protein 2, also known as RING1B), play an important role in regulating germ-cell-specific gene expression both in the male [69] and the female germline [70]. The two paralog catalytic subunits act redundantly [117], but it is still possible to study their function in germ cell differentiation by using a germ-line specific double mutant knockout lines. *Ring1^-/-^* mice are viable and fertile [118], but *Rnf2^-/-^* mice are embryonic lethal [119], therefore the germline-specific functions of the double mutant are not feasible to analyze directly. In order to examine the germ-cell-specific functions of the two RING paralogs, conditional deletion of *Rnf2* was generated on *Ring1^-/-^* background [69], using a *Ddx4-Cre* driver, which is expressed in germ cells after E14 (referred as PRC1cKO) [120]. The germline double mutant mice of RING1 and RNF2 units represent a complete loss of catalytic PRC1 function. PRC1cKO males are all infertile and have small testes [69]. The number of undifferentiated spermatogonia is normal at birth in the PRC1cKO but is slowly decreasing with the mutant animal’s age. In parallel, the differentiating meiotic cells, marked by γH2A are largely diminishing, indicating that the mutant spermatogonia are not able to pass the mid-pachytene stage. The cell cycle and proliferation rate of the PRC1cKO spermatogonia seem to be normal, but apoptosis is increased in the testis of the PRC1cKO animals [69].

PCGF6 and RING1 form the E3 ligase catalytic core of the ncPRC1.6 complex by directly interacting with each other. Although ncPRC1.6 regulates germ-cell-specific and meiotic genes, the genomic binding sites of PCGF6 and RING1 overlap only partially in ES cells, according to the published Chromatin Immune Precipitation followed by whole genome sequencing (ChIP-seq) experiments (Geo accession: RING1B-GSM1041372, PCGF6-GSE84905, RYBP-GSM1041375). Moreover, if the common ChIP binding sites of PCGF6 and RING1 are taken and compared to the sites occupied by RYBP [121], the number of common ChIP targets is further reduced (Figure 2B).

It is well demonstrated that RYBP, RING1, and PCGF6 are ncPRC1.6 members, and all three proteins regulate a discrete set of germ-cell-specific target genes in ES cells, including the ones associated with meiotic functions [104], although none of the three presumptive partners are exclusively associated with each other. This explains the differences in their binding pattern obtained by ChIP-seq experiments. First RING1 can associate with chromobox (CBX) proteins in canonical PRC1 complexes, which contain neither RYBP nor YAF2 subunits [83]. According to chromatin immunoprecipitation data, RING1 containing canonical CBX containing and non-canonical RYBP/YAF2 containing complexes are able to regulate highly different sets of targets [121]. Moreover, RING1 plays a direct role in activation on a subset of its regulatory targets, which are targeted independently of CpG methylation or ncPRC1.6 specific DNA binding subunits. It was found that RING1 bound to the TSS of many genes that are normally active in the germline. Recent studies have demonstrated that the interaction with Spalt like transcription factor 4 (SALL4), a specific transcription factor that activates germ cell genes [122], is also necessary for the activator function of RING1 on this special subset of germ-cell-specific targets. For example RING1 is found near to TSS of *Tdrk* and *Mael* together with SALL4 in undifferentiated spermatogonia in the testes and promotes their activation [69]. *Tdrk* and *Mael* are, in turn, found to be repressed by PCGF6 containing complexes in ES cells [104] suggesting that the same regulatory regions and factors can contribute to opposing functions in the course of development bound by a different set of available regulators. *Pcgf6* is expressed abundantly in ES cells, and during germ cell differentiation but it is hardly detectable in E15.5 testis according to comparative in situ RNA hybridization studies, published in GXD database (http://www.informatics.jax.org/assay/MGI:5539656#GUDMAP_12114_id) [123], while SALL4 shows strong germline-specific expression during early development [122], which becomes restricted to the gonads after birth [124]. This gene expression pattern enables the possible substitution of RING1 containing ncPRC1.6 complexes repressing germ-cell-specific genes with SALL4-RING1 containing forms able to transcriptionally activate the same loci during germ cell differentiation.

RYBP can be substituted with YAF2 in PCGF6 containing ncPRC1.6. complexes [83], but the two proteins are not simply interchangeable. It is well demonstrated that regulatory targets of RYBP and YAF2 differ profoundly [125]. The significantly upregulated gene sets of the transcriptome of the *Rybp^-^* and *Yaf2* mutant ES cells are also dissimilar (Figure 3A), even in experiments performed and analyzed in the same laboratory in parallel, indicating non-redundant functions of the two homologs [125]. RYBP can also interact with ncPRC1.6 independent partners, like OCT4/POU5F1 transcription factor, contributing to gene activation instead of repression. The best known example of this type of regulation has been described by Li and coworkers on the *Kdm2b* promoter [71], where OCT4 and RYBP are both necessary for proper activation. Taken together, the diverse regulatory interactions of the three partner proteins can explain the considerable differences in binding sites identified by ChIP-seq experiments and indicates the importance of examining the specific functions of all ncPRC1.6 members independently in germ cell differentiation and transcriptional regulation.

## 4. Interactions of BMP and RA Signalization Pathways and PRC Complexes in Germ-Cell-Specific Regulatory Processes

Several studies demonstrated that ncPRC1 complexes affect the BMP pathway. Loss of RNF2, a catalytic subunit of all PRC1 complexes, results in the upregulation of several BMP pathway members [127]. PCGF2, the ncPRC1.2 subunit represses negative regulators of the BMP pathway [128,129], while the absence of MGA results reduced BMP signaling during embryonic development in zebrafish [130,131].

Other studies pinpointed the connection between the RA and BMP signaling pathways during germ cell differentiation. Studies using SSCs [132] (Figure 1) and vitamin A deficient (VAD) mice [133] proved that during spermatogenesis BMP4 and RA signaling act cooperatively and induce transcription of *Stra8* and *c-Kit* through SMAD1,5,8 activation. During oogenesis RA and BMP signaling acts together and repress early PGC genes and upregulate late germ-cell and fetal oocyte genes, including *Stra8*. Taken together RA and BMP signaling cooperate in pushing PGCs toward female fate [134]. *Stra8* is not only a target of RA signaling but also a direct target of PRC1 and PRC2 complexes in PGCs, which maintain *Stra8* repression [70]. All of these studies demonstrate that the interacting RA and BMP pathways are influenced by PRCs in the course of PGC differentiation.

## 5. Connections between RYBP and Germ Cell Fate Regulation

RYBP is a 228 amino acid long, evolutionarily conserved Zn-finger protein. In many species like in *Drosophila melanogaster*, *Mus musculus*, and in humans, a homolog of RYBP is found, called YAF2 [135]. The two homolog proteins show the highest match in their N-terminal Zn-finger motifs and another block of a less similar region is present at their C-terminal end, while the two conserved regions are separated with a long non-homologous stretch [87,88]. RYBP and YAF2 are able to interact with a similar set of partner proteins [83,87,135], but in many cases, they have distinct functions [88,125]. The expression of the two homologs also differ in preimplantation stages. The maternally inherited YAF2 is often localized in the pericentromeric regions of the zygote. RYBP is absent in this stage but strongly induced from the 4-cell stage onward, while YAF2 protein is present only in these early stages but becomes almost undetectable after the 4-cell stage in mouse embryos [136]. *Rybp* is an essential gene, *Rybp^-/-^* mouse embryos die soon after implantation (E5.5–E6) [137], proving that *Rybp* has an essential role during early embryonic development, which cannot be compensated by the presence of *Yaf2*. There is no published data about the phenotype of *Yaf2* knockout (*Yaf2^-/-^*) mice yet, hindering attempts to study its in vivo role during embryonic or germ cell development. It is also challenging to study the role of *Rybp* in germ cell development, as the lethal phase of *Rybp^-/-^* mice proceeds the specification of germ cells (E6.25–6.5) [37,138,139]. However, homozygous *Rybp^-/-^* ES cells are viable, thus “luckily” it is possible to study the role of *Rybp* during germ cell development in vitro in ES cell-based model systems. This can be achieved, due to the fact that *Rybp^-/-^* ES cells proliferate normally [126,140,141] as opposed to *Yaf2^-/-^* ES cells, which have compromised proliferation, due to their altered cell cycle profile. The non-overlapping mutant phenotypes of *Rybp^-/-^* and *Yaf2^-/-^* ES cells also indicate non-redundant functions of the two homologs [125].

The possible role of *Rybp* in germ cell specification can be studied using the viable and well proliferating *Rybp^-/-^* ES cells. *Rybp* mutant ES cells were produced by different methods and whole-genome transcriptome data of wild type and *Rybp* mutant ES cells were compared in independent experiments [94,125,126]. By analyzing the raw reads, we made a list of 2128 upregulated genes (log_2_ fold change ≥ 2) from Ujhelly et al. [126], where a targeted deletion of RYBP [137] was introduced in wild type R1 ES cell line. We found 819 significantly upregulated genes from the data-sets published by Rose et al. [94] where a conditional knockout *Rybp* mutant allele in R1 ES cells was used, while 609 significantly upregulated genes were identified in the *Rybp* knockout by Zhao and colleagues using mouse V6.5 ES cell line [125] (*See Methods*). Although the genetic background and the culturing conditions were not identical in the different experiments, the authors of the three publications all agree that the most striking difference between the transcriptome of wild type and *Rybp^-/-^* ES cells corresponds to germline specification and meiotic processes [94,125,126,140]. Analyzing the similarities between these lists of upregulated gene sets we identified 65 genes that were commonly upregulated more than two-fold in all three studies (Appendix A) (Figure 3B). By utilizing the Gene onthology (GO) term enrichment analysis (Figure 3C), the commonly upregulated gene set in the *Rybp* mutant ES cells showed clear connection to germ cell development and meiotic processes (Figure 3C) (Appendix A) suggesting that *Rybp* have a vital role in repressing genes required for germ cell specification in the somatic lineage.

There are important germ cell and germ granule components, such as *Ddx4*, *Dazl*, *Mael*, *Piwil2*; key transcriptional (such as *Prdm14)* or meiotic (such as *Mei1*, *Sycp1*, *Sycp2*, *Mlh1* and *Meioc*; male-specific *Ift88*, *Hspa2*, *Foxo)* regulators of germ cell specification are all in the upregulated pool in the whole genome transcriptome dataset of Ujhelly et al. [126]. When raw mRNA counts, (Reads Per Kilobase Million, RPKM), from wild type and *Rybp^-/-^* cells were plotted into a scatter plot (Figure 3E), the trend-line of the wild type-mutant ratio of germ cell genes transcription runs higher than the general trend-line calculated for all genes analyzed, showing that the germ cell genes are more upregulated compared to the whole gene set in the *Rybp^-/-^* cells (Figure 3D,E).

Regulation of the germ-cell-specific transcriptional program has not been described in detail in vivo in the model organisms yet. We do not know the exact amount of the important germ-cell-specific mRNAs in different developmental stages. Sensitive new methods of single-cell qRT-PCR revealed significant transcriptional heterogeneity amongst developing germ cells in vivo [142]. There are significant differences even in the transcriptome of well-defined cell types [142]. Heterogeneity in the transcription pattern might be a fundamental feature of the germline at the earliest stage of germ cell specification to the initiation of spermatogenesis from SSCs (FIG1) [142]. This might be part of the reason why the level of germ-cell-specific transcripts varies in vitro in cultured wild type and mutant ES cell as well.

Although the YAF2 protein have many common binding partners with RYBP, like polycomb type repressor subunits of PRC1 complexes [83] and YY1 [135], there is no significant overlap in the set of the more than 1.5 times upregulated genes in *Rybp* or *Yaf2* mutant ES cells in the transcriptome data obtained and analyzed in the same laboratory (Figure 3A) [125]. Germ-cell-specific gene expression is only upregulated in the *Rybp* mutant ES cells in all independent studies [94,125], but not in *Yaf2* mutant cell. Not all germ cells specific genes are repressed by RYBP and upregulated in *Rybp^-/-^* ES cells. Some loci behave on the contrary (Figure 3E). Certain germ-cell-specific genes are significantly downregulated in the mutant background like *Dnd1*, *Bmp8b*, *Prdm1* represented by the dots under the trend-line (Figure 3E). It seems that the alteration of the transcription of germ cell genes is a specific function of RYBP. Future conditional mouse models or ES cell-based in vitro differentiation systems need to clarify the role and possible redundancies between *Rybp* or *Yaf2* during normal germ cell differentiation.

Germ-cell-specific gene expression is repressed in the somatic cells, accordingly, *Rybp* is expressed in most somatic tissues [126,137,141]. *Rybp^-/-^* embryos do not survive to the stage of initiation of PGC determination [137], but RYBP protein is readily detectable in PGCs of the wild type egg-cylinder stage mouse embryos underlining its role in germ cell development [137]. RYBP is also present in the testis of adult mice (Figure 3F). In order to check the tissue localization of RYBP, cross-sections of C57BL/6 adult mouse testis were stained with hematoxylin and eosin and anti-RYBP antibody (Merck Millipore, Cat.No AB3637, Darmstadt, Germany). High level of RYBP is present in the proliferating spermatogonia (Figure 3F). As differentiation proceeds, the level of RYBP protein is reducing, but RYBP never disappears from the differentiating male germ cells. A similar tendency is visible at in situ RNA hybridization of E15.5 testis, published in the GXD database (http://www.informatics.jax.org/assay/MGI:5540282#GUDMAP_6516_id) [123] (Figure 3F). Key interacting partners like PCGF6 show different RNA expression patterns in the testis. Consistent with the role of PCGF6 containing ncPRC1.6 complexes repressing the germ-cell-specific transcriptional program in the soma. The level of *Pcgf6* RNA is very low in adult testis, where the late germ cell program is activated, and repression of many *Pcgf6* targets is not necessary anymore. The maintained expression of RYBP at the same time might indicate the need for some ncPRC1.6 independent functions of RYBP in regulating late germ cell specific events, like meiotic processes independently of ncPRC1.6.

## 6. Commonalities between Germ Cells and ES Cells

The relation between somatic, germline and ES cells is still puzzling. Most ES cells are produced by culturing ICM or primitive ectoderm (PE) cells of preimplantation embryos. The culturing of the isolated ICM/epiblast cells does not always lead to a successful derivation of ES cells in the Petri dish, even in optimal conditions. By tracking the expression of Octamer binding transcription factor 4 (OCT4/POU5F1), a marker of pluripotency in the cultured ICM and PE cells, it turned out that OCT4 expression is retained only in a minority of the outgrowing cell population [143]. This finding indicates that as ES cells are selected from ICM or PE cells, they have to adapt to the new artificial in vitro environment and only a small subset of the isolated primary cells are able to do that. By the time the isolated embryonic cells gain the capacity of indefinite multiplication, their original identity changes. The transcriptional pattern of ES cells and ICM cells differs. The gene expression pattern of ES cells most resemble to that of early PGCs, suggesting that ES cells are most similar to embryonic germ (EG) cells [144,145]. Independent findings support this hypothesis. Activation of the germ-cell-specific transcriptional program promotes the derivation of ES cells. *Prdm1/Blimp1* is a key transcriptional repressor of the somatic program during germ cell specification [146]. Activation of *Prdm1/Blimp1* in ICM cells is a strong indicator of their high competence to become ES cells. Prospective sorting of BLIMP1 positive cells increased ES cell derivation efficiency by 9-fold [145]. Knowing that activity of germ-cell-specific transcription factors, like *Prdm1*, promotes ES cell derivation, it is not surprising that substantial number of early germ cell markers are also expressed in the undifferentiated ES cells. Not only core pluripotency network members are transcribed, which are also important for maintaining the pluripotency of germ cells, like *Oct4/Pou5f1* and *Nanog*, but germ-cell-specific marker genes like *Kit, Dazl*, and *Dppa3* are also active in most ES cells. This fact can be especially puzzling, when ES cells are differentiated towards germ cell specificity. The course of germ differentiation from ES cells really hard to follow, because regularly used germ-cell-specific markers are often detected in the undifferentiated state as well [147,148]. In turn, the late markers of germ cell differentiation, like *Ddx4* are typically undetectable in human ES cells [149] indicating that the low-level transcription of the markers found is not merely the result of leaky transcription characteristic of stem cell state. ES cells are not just a simple equivalent of early germ cells.

In spite of the similarities in the transcriptional pattern, ES cells and germ cells differ in many respects. PGCs are not able to self-renew forever as ES cells do, PGCs are destined to maturate, migrate and finally differentiate to a sperm or an egg [1,9] and normally cannot differentiate towards somatic lineages any longer [150]. On the other hand, ES cells are not only pluripotent and can differentiate to all three germ layers but can self-replicate for a prolonged period of time. It is well known though, at the same time, ES cells have a capacity to become germ cells, when they are introduced into a proper host embryo. They are “germline competent”. It means that ES cells can be inserted into the developing gonads in the so-called “chimera mice”. The efficiency of germ cell competence of ES cells of different origins is highly variable and depends on many factors [151,152]. The preserved germline competence of ES cells indicates that the germline-soma decision has not been made in cultured ES cells yet.

There is another, almost forgotten, historical connection between ES cells and germ cells: germline teratomas. In the 70s, mouse teratocarcinomas paved the way for the derivation of ES cells [153]. Teratocarcinomas are germ cell tumors, containing both differentiated tissues and stem cells called embryonal carcinoma (EC) cells [154]. The EC component of the tumor ensures that it can be serially transplanted in mice [155], can be cultured and differentiated in vitro for a prolonged period of time [156]. EG cells can arise from embryonic teratoma (ET) cells, which can never be converted into somatic cells again [150].

It is worth to notice that in species where teratocarcinomas occur at a clinically significant frequency, such as in mice and humans, the derivation of ES cells is relatively easy, while in species where this condition is exceedingly rare, like in rat, ES cells are more difficult to derive [144]. A similar correlation can be found even within the same species. In the inbred line 129 of mice, where teratomas occur at very high frequency [157], establishing a new ES cell line is easier [151,158] than from other inbred lines with a different genetic background [152]. Matsui and colleagues reported the successful isolation of a new type of pluripotent stem cells, which can be differentiated to all three germ layers from the nascent teratoma of *Dead end 1* (*Dnd1*) mutant [159]. *Dnd1* mutation is present in 129 strain and greatly enhances teratoma formation on its own. The isolated teratoma forming stem cells harboring *Dnd1* mutation can be cultured in the presence of leukemia inhibitory factor (LIF). They are morphologically similar to ES cells, express *Sox2*, *Nanog* and *Oct4/Pou5f1*, and their transcriptome is indicating that they are in between the naive and primed pluripotency stage [160]. It seems that EG cells in neonatal testes of *Dnd1* mutants cross this crucial intermediate stage, which might be a prerequisite of their ES-cell-like properties [159].

These notions support the hypothesis that the expression of a subset of germ-cell-specific genes is normal and inevitable in ES cells. Germ cell genes are generally silenced in normal ICM and PE cells in vivo, but the activation of some germ-cell-specific genes might be essential for the maintenance of the pluripotent state for a prolonged period of time [144]. Probably this is the reason, why the same subset of germ-cell-specific genes is often active in tumor cells as well in different invasive cancers [4]. Combinations of germ cell pluripotency factors like *Nanog*, *Oct4/Pou5f1,* and *Sall4* are often used as tumor markers [161,162,163], and some authors even suggest that there is a special connection between induction of meiosis and oncogenesis [164].

## 7. RYBP Is Connected to Germline Specific Functions by a Plethora of its Binding Partners

### 7.1. Natively Unfolded Structure of the RYBP Enables It to Affect Germ-Cell-Specific Functions in Multiple Ways

RYBP is a multifunctional protein, it has numerous known binding partners, and it performs many autonomous and often seemingly unrelated functions, without portioning these functions into different domains of the protein. Based on these findings, RYBP undoubtedly belongs to the family of moonlighting proteins [165]. RYBP is natively unfolded and it is able to change its tertiary structure upon binding to different partners [166]. According to that the crystal structure of native full-length protein has not been resolved, only a small C-terminal region of RYBP (145-198 amino acid (AA)) was crystallized together with RING1 ubiquitin ligase [167], but even this small portion of RYBP is unfolded in the absence of RING1, and the final interacting beta-sheet is only formed upon binding [167]. The moonlighting characteristics of RYBP can explain its interaction with many, profoundly different binding partners, which are often connected to different aspects of germ cell specification, repression or activation of germ-cell-specific genes, programmed cell death, protein ubiquitination, nuclear transfer reprogramming, meiotic processes, and DNA repair.

### 7.2. RYBP is Connected to Germ-Cell-Specific Apoptotic Processes via FANK1

RYBP was first named as DEDAF (Death effector domain-associated factor), because it was identified as a DED (Death effector domain) interacting factor functioning in apoptotic processes both in the nucleus and in the cytoplasm [168]. Later, it was revealed that RYBP is able to interact with other factors without the DED domain also regulating programmed cell death. Fibronectin type 3 and ankyrin repeat domains 1 protein (FANK1) is one of them [169]. The FANK1-RYBP interaction is mediated by an FNIII domain of FANK1 and the C-terminal serine/threonine rich domain of RYBP. The binding of RYBP stabilizes FANK1 by inhibiting its proteasome-mediated degradation. FANK1 is a conserved apoptotic protein highly expressed in the testis [170]. The FANK1 protein not only exclusively expressed in the testis, but its expression is restricted to the mid-late pachytene stage of spermatid differentiation [171]. FANK1 is also highly expressed in germinal vesicle (GV) oocytes [172]. The GV is the specific name for the nucleus of an oocyte that is arrested in prophase of meiosisI. The nuclear envelope of the GV breaks down in response to hormonal stimuli, when the oocyte finally resumes meiosis prior to ovulation. In this stage, in the female germline, FANK1 might have a similar function as in pachytene spermatids, it safeguards late meiotic processes and if the quality of DNA is compromised, the cells are apoptotized instead of commitment to ovulation FANK1 is localized in the nucleus and GO analysis shows that FANK1 might have DNA binding capacity and may also function as a specific transcription factor regulating the transition from meiotic to haploid phase of spermatid differentiation [171], later the specific DNA binding capacity of FANK1 was detected, and its presumptive regulatory targets were identified [173]. The RYBP-mediated stabilization of FANK1 might be important for the proper regulation of apoptosis during the course of meiosis, and this interaction connects RYBP with germ-line-specific apoptotic processes.

The developmental-stage-specific functions of the two well defined apoptotic proteins, FANK1 and RYBP, in germ cells support the notion that many proteins that have been specifically implicated in apoptotic processes might have vital, phylogenetically conserved, and sometimes even apoptosis-unrelated functions during development [174]. It might be possible that the cell-death-unrelated functions of some apoptotic proteins can be more ancestral, like in adaptation of environmental stress and DNA damage and become potential death effectors only later, during evolution [174]. Both RYBP and FANK1 proteins can be a good example of this concept. 

### 7.3. RYBP Can Contribute to the Regulation of Germ-Cell-Specific Gene Expression Directly as a Subunit of ncPRC1.6

Since RYBP has been purified in all ncPRC1 complexes [83,100], and it is best known as a PcG member, controlling lineage-specific transcription and modulating differentiation capacity of ES cells [121,126,141,175,176,177].

RYBP is named after two of its binding partners RING1 and YY1 [87], which are both key members of the PcG epigenetic repressor proteins [178,179] and involved in the maintenance of the epigenetically repressed states of homeotic genes by compacting chromatin [180,181]. The first name giving binding partners of RYBP, RING1, is the catalytic E3 ligase subunit of all PRC1 complexes as well as its paralog RNF2, which is also a binding partner of RYBP [182]. The RAWUL domain [102] of RING1, which can mediate the homo-dimerization of the protein [183,184] and also needed for effective RING1-CBX interaction in canonical PRC1 complexes [184] mediates RING1-RYBP interaction in the ncPRC1 complexes too [167]. This finding indicates that RING1 can be stabilized without its canonical or non-canonical binding partners as a dimer or can be modularly associate with different PRC1 type complexes. The fact that RYBP occupies the exact same binding surface on RING1 as POLYCOMB (PC) homolog CBX proteins reassures that RYBP containing non-canonical and CBX containing canonical PRC1 complexes are always assembled independently. Mosaic, canonical/non-canonical, PRC1 complexes never form. Although the C-box domain of CBX7 and the RING binding region of RYBP form a nearly identical intramolecular beta-sheet, the loop structure and the amino acid sequence of RYBP and CBX7 differ profoundly. RYBP and CBX7 attain altered conformations upon binding to RNF2 [167]. The basically unstructured nature of RYBP is a prerequisite for this interaction and it makes possible that the portion of RYBP important for recognizing RING1 can also interact effectively with many other binding partners harboring an ubiquitin-like fold.

YY1, the second PcG member able to bind RYBP, codes for a sequence specific DNA binding protein and its fly homologs PHO and PHOL play a pivotal role in the targeting of polycomb complexes [185,186]. *Yy1* is an essential gene in mice. Both the *Yy1^-/-^* and the *Rybp^-/-^* show peri-implantation lethality [187]. Sequence-specific PHO/YY1 binding sites were described [188] and found both in drosophila Polycomb Response Elements (PREs) [189] and in mammalian homeotic regulator genes [190,191]. PHO protein, the fruit fly homolog of YY1 plays an important role in vivo targeting PRC complexes. PHO/YY1 binding sites are regularly found in Drosophila PREs. Many cases were described, when mutation of the conserved PHO/YY1 binding sites abolishes PRC binding and repression of target genes by PcG system in Drosophila [192,193]. The importance of PHO/YY1 mediated targeting of PcG repressor were clearly demonstrated both in reporter constructs and [194,195,196,197] in native genomic location, in regulatory regions of homeotic genes [198]. Later it was found that, contrary to Drosophila, YY1 binding sites in humans are mainly located at active promoters and missing from most classical PcG target genes. It was supposed that interactions involved in targeting PRCs to YY1 sites are somehow lost in the mammalian lineage [186]. Mammalian RYBP protein can bind RING1 and YY1 parallel, and therefore, it can serve as a bridging factor between them.

RYBP interaction with YY1 does not always lead to repression. RYBP can connect YY1 binding sites with transcriptional activation. Gene activation is supposed to be the main function of YY1 in mammals, suggested by a TSS proximity of binding sites. It is clearly demonstrated in the case of the Cell division cycle 6 (*Cdc6*) promoter. CDC6 functions as a protein regulating early phases of DNA replication. There are adjacent E2F and YY1 binding sites at the *Cdc6* regulatory region, which are both required for promoter activity. YY1 and RYBP in combination with either E2F2 or E2F3 can stimulate the activity of the *Cdc6* promoter synergistically, dependent on the marked box domain of E2Fs (Figure 4). RYBP can interact with E2F2 and E2F3 directly, simultaneously with YY1 (Figure 4A), and associated with the *Cdc6* promoter at the G_1_/S phase, when the gene is active. This interaction is specific because E2F1 binding is not detectable [89]. As demonstrated by these findings, RYBP can function as a bridging factor interacting with multiple DNA binding transcription factors besides YY1 and mediate both sequence-specific activation or repression of target genes [199,200].

This ability of RYBP is not limited to E2F2/E2F3 interaction. RYBP is able to interact with another E2F protein, E2F6, which has a similar DNA binding ability, but it is generally associated with repression (Figure 4). E2F6 belongs to the PcG and purified in many ncPRC1.6 type complexes [83,90,202]. The C-terminal 144-228 AA of RYBP is important to RYBP-E2F6 interaction, while in E2F6 the C-terminal 179-240 AA region is necessary [90]. The C-terminal of E2F6 harboring the dimerization and marked box domain is proved to be important for the interaction [90] similarly to E2F2 and E2F3 [89]. E2F6 acts as a repressor of certain male germline-specific genes [111]. E2F6 can contribute to the repression of the germ-cell-specific transcriptional program in two independent ways. First it can act as a classical PcG protein, being a member of ncPRC1.6 and profoundly contributing its targeting [92], but it can also interact with DNA methyl-transferase 3b (DNMT3B) which contribute to the repression of several germlines, especially meiotic genes, in somatic tissues by methylating their promoter regions (Figure 4C). The recruitment of DNMT3B depends on E2F6, and E2F6 binding sites are the common hallmarks of DNMT3B regulated promoters [201]. Based on its interaction with E2F6 RYBP can contribute to the silencing of meiotic genes as a member of PRC1 mediating H2A 119 ubiquitylation and chromatin compaction, and beside this, it can function by enhancing site-specific DNA methylation of E2F6 targets and may even play a role in coordinating these processes.

### 7.4. RYBP Regulates Germ-Cell-Specific Gene Expression Indirectly through its Multiple Connection with the Ubiquitin System

RYBP is closely connected to the ubiquitin system in many ways. RYBP is able to bind ubiquitin and ubiquitinated proteins, and it can be ubiquitinated itself [175,203]. RYBP binds directly to RING1 protein [87], which is a catalytically active auto-inhibited E3 ubiquitin ligase [182], able to specifically ubiquitilate H2A at K119 [79,204,205] with a well-defined Npl4 zinc finger (NZF)/RanBP2 zinc finger domain, destined for interaction with ubiquitin and ubiquitinated proteins [206]. The region of the RING1 protein, which is bound by the RYBP NZF domain has a ubiquitin-like structure, a so-called ubiquitin fold [184]. Binding of RYBP greatly increases the catalytic activity of all RING containing complexes analyzed as compared to CBX containing complexes [83,94,175]. The mechanism of this enhancement is not clarified in all aspects, but RYBP acts partially by altering the conformation of RING1 in the different ncPRC1 complexes [94] and beside this stabilizing RING1 itself by blocking its ubiquitin-dependent degradation [207]. The mechanism of the alleviation of the proteasomal degradation of RING1 has only been revealed recently. RYBP can bind Ubiquitin protein ligase 3A (UBE3A) another specific E3 ubiquitin ligase, and its binding promotes its poly-ubiquitination and consequent proteasomal degradation. The reduction of the UBE3A level lengthens the half-life of RNF2 by alleviating poly-ubiquitination dependent proteasomal degradation, thus stabilizing PRC1 complexes and enhancing their catalytic activity by this way too [207]. It is well known that PRC1 complexes, especially ncPRC1.6 play an important role in repressing germ-cell-specific gene expression in the somatic tissues [104]. The connection between the stability of RNF2 can be easily correlated with avoidance of ectopic germ cell gene expression and consecutive entry to meiotic prophase in somatic cells. Surprisingly RNF2 is also required in female PGCs to maintain a high level of *Nanog* and *Oct4*/*Pou5f1* expression [70], therefore it is an important factor for maintaining the pluripotency network active in early germ cells [208] and directly regulating *Sall4* [69].

The latest identified ability of RYBP is to recognize and bind K63 ubiquitin chains and contribute to double-strand break repair [209]. This finding opened the whole new avenue for research. This finding raised the possibility that DNA repair mediating activity of RYBP can contribute to the organization of homologous recombination repair at double-strand brakes during crossing over as well.

### 7.5. RYBP Directly Interacts with OCT4/POU5F1 and Can Be Targeted to Pluripotency Target Genes Bound by OCT4/POU5F1

Germ cells express numerous pluripotency markers such as OCT4/POU5F1, FUT4/SSEA1, KLF4, NANOG [14]. Changes in their expression can dramatically affect germ cell fate. RYBP has been connected directly to the core pluripotency network. *Rybp* is directly regulated by OCT4/POU5F1, and it has been identified as an OCT4 binding partner with mass spectrometry analysis together with RNF2 [210]. The RYBP-OCT4 interaction was validated in an independent study, where it was proven that the depletion of *Rybp* reduces OCT4/POU5F1 binding on *Kdm2b* promoter, reducing its transcriptional activity (Figure 4E). RYBP can, therefore, not only physically interact with OCT4, but facilitate its binding to target promoters and play a direct role in the transcriptional activation process on the targets [71].

### 7.6. Loss of Rybp Affects the RA Signalization Pathway and the Level of Nanog Simultaneously in ES Cells

Embryonic development of germ cells and reproductive capacity of both males and females is highly affected by the RA pathway [211]. RA is required for the initiation of meiosis in both male and female mammals. The amount of RA and effectivity of RA signaling depends on RA synthesis, degradation and also on the presence and expression level of the RA receptors (RARs). RA is used as an inducer of in vitro differentiation assays in ES cell-based systems and used for the differentiation of PGC like cells [148,149,212,213,214,215,216,217,218,219]. RA induces the expression of many germ-cell-specific genes genome-wide [220] and RA signalization plays a key role in regulating meiotic entry too [221]. This effect is mediated by several RA responsive target genes, like *Stimulated by retinoic acid gene 8* (*Stra8*), an essential factor of meiotic entry [222].

Many RA pathway members (e.g., *Cellular retinoic acid binding protein 1*
*(Crabp1)*, *Retinol dehydrogenase 14 (Rdh14**))* are upregulated to some extent in *Rybp^-/-^* ES cells according to the transcriptome data of Ujhelly et al., 2015 [126] (Table 1). This raises the possibility that RA level and consequent signalization processes are also altered in the *Rybp^-/-^* background [126].

Surprisingly RYBP is detected on the regulatory region of only one upregulated RA pathway member, *Crabp1* according to ChIP-seq data (Table 2) [121]. Three RA receptors - *Retinoic acid receptor beta* (*Rarb)*, *Retinoid X receptor alpha* (*Rxra),* and *Retinoid X receptor gamma* (*Rxrg)* -, and Crabp1 are bound by RNF2 [121], while PCGF6 is only detected on the *Rxrg* promoter [109]. The question arises, if RYBP does not seem to regulate these targets directly, what causes their derepression in the *Rybp^-/-^* ES cells? The existence of a possible auto-regulatory loop in RA signalization was proposed previously [223]. From the seven upregulated RA signalization factors, four genes, *Adh4*, *Rarb*, *Rxra,* and *Rxrg*, are regulated by RA signalization directly in *Rybp^-/-^* ES cells.

Besides the activation of the RA pathway, one of the key pluripotency genes, *Nanog* was found to be upregulated in the *Rybp^-/-^* ES cells (Figure 3D). NANOG is a homeodomain containing transcription factor playing an important role in maintaining ground state pluripotency by activating a large set of target genes [224]. The paternal X chromosome is still repressed in early blastocyst cells in female embryos, and remain repressed in trophoblast cells all through their development, but NANOG positive ICM cells are able to reactivate the paternal X chromosome in female embryos after E3.5, which enables subsequent random inactivation of the two X chromosomes in somatic cells. NANOG is expressed in a subset of ICM cells, which eventually give rise to the pluripotent epiblast at E4.5. NANOG is often called the gatekeeper of the pluripotent ground state [225]. Ectopic expression of NANOG alone can induce germ cell fate independently from BMP4 signaling by coordinately switching on many germ cell regulatory proteins [61]. Specific expression of NANOG is detected in PGCs in fetal ovaries in humans [226] and in differentiated pachytene spermatocytes in adult testes suggesting that it might have a specific function in regulating meiosis [227]. The fact that *Nanog* is upregulated in the *Rybp^-/-^* ES cells (Figure 3B, based on [126]) raises the possibility that the upregulation of a subset of germ-cell-specific genes are not only the mere consequence of ncPRC1.6 dependent derepression in the lack of RYBP, but resulted from an over-activation of a NANOG based regulatory network, important for germ cell determination. Although sustained expression of *Nanog* alone can induce germ cell fate but the exact mechanism of this phenomenon had not been analyzed transcriptomically in mice. Murakami and colleagues described the key targets of NANOG regulation [61]. Amongst these *Prdm14* (1.6 times), *Dppa3* (1.91 times), and *Dazl* (14 times) upregulated in *Rybp^-/-^* ES cells according to the dataset of Ujhelly et al., 2015 [126]. Two out of these three targets are under the 2 times upregulation limit, therefore not found in the 2 times upregulated datasets of 65 genes we were using. Unfortunately, genome-wide NANOG targets are only determined in human ES cells [228]. We found no overlap between the 65 common targets, but 81 genes are common (Appendix A) between the 2117 upregulated genes in Ujhelly et al., 2015 [126] and the 1244 identified ChIP binding sites of NANOG identified [228]. Out of these 81 genes 4 connected to germ cell development according to the GO database (Appendix A).

Germ-cell-specific factors generally found to be upregulated in the *Rybp^-/-^* background from the three independent studies [94,125,126]. GO studies showed that 18 genes are key germ cell marker genes out of the 65 commonly upregulated genes. As meiosis plays a central role in germ cell determination, we next verified the presence of the meiotic marker genes in more than one published transcriptome. Gene ontology studies showed that 26 genes attributed to meiosis upregulated in the *Rybp* mutant ES cells in more than one independent transcriptome data (Table 2). We checked the presence of published ChIP-seq positive sites of RING1 (PRC1 and ncPRC1 core member), RYBP (ncPRC1 core member) [121], PCGF6 (ncPRC1.6 core member) [109] and RAR (RA signaling pathway members) [229] to specify if the upregulation of meiotic genes are a direct consequence of the compromised ncPRC1.6 mediated regulation, or the RA pathway (Table 2). The promoter region of many genes expressed in germ cells and upregulated in *Rybp* mutant ES cells contain all three ncPRC1.6 proteins (RYBP, RING1, PCGF6) (Geo accession: RING1B-GSM1041372, PCGF6-GSE84905, RYBP-GSM1041375) suggesting that germ cell developmental genes are often regulated by ncPRC1.6. in ES cells too. These genes including *Ddx4*, *Dazl*, *Piwil2*, *Sycp1*, *Sycp2*, *Smc1*, *Cpeb1*, *Gm1564*, *Tex11*, *Mael*, *Tdrkh*, *Bol1* and *Mov1* are probably repressed directly by RYBP containing ncPRC1.6 in wild type ES cells. Their elevated transcription in the mutant background is probably the direct consequence of the impaired formation of ncPRC1.6 complexes in the lack of *Rybp*. Some of the upregulated targets harbor RYBP and RING1 sites only, like *Sycp3*, and *Tex15*, but they lack PCGF6 binding sites in their promoter regions. These genes are probably regulated by different ncPRC1 complexes (ncPRC1.1, ncPRC1.2/4 or ncPRC1.3/5). There is only RING1 binding site in the *Prdm14* gene promoter, which makes it feasible that canonical PRC1 regulates this gene. This notion is supported by the fact that *Prdm14* is upregulated in *Cbx6^-/-^* background too [230]. However, only RYBP binds to *Hormad2,* the gene essential for the meiotic prophase [231]. 

Some of the upregulated targets, like the RA inducible *Stra8*, which is the member of the X linked lymphocyte regulated (*Xlr*) family and the Argonaute family member *Piwil4,* are not bound by any PRC1 members at all. *Stra8* harbors functional RA response element (RARE) in its promoter region [232] and as expected, *Stra8* is a positive hit in the RAR ChIP-seq dataset. Similarly, to *Stra8*, RAR also seems to bind to the *Piwil4* and *Hormad2* promoter in ES cells. The upregulation of the *Xlr* genes is even more enigmatic. The *Xlr* family has been originally linked to immune functions and has been found to be expressed in lymphocytes [233]. Later it was described that the *Xlr* genes are also expressed in mouse oocytes during early meiotic phases [234]. As *Xlr* promoters do not seem to be repressed neither by canonical-, or non-canonical PRC1 complexes nor RARs, their upregulation is probably not a result of mere derepression, but a direct consequence of activation by some meiosis inducing transcription factor, which is de-repressed in the *Rybp^-/-^* ES cells.

### 7.7. RYBP and DDX5 is Connected to Reprogramming via a microrna Based Regulatory Loop

The interaction of RYBP with RNA binding proteins like DEAD box polypeptide 5 (DDX5) has been previously revealed [71]. The interaction of RYBP and DDX5 not just contributes to the inhibition of the entire reprogramming processes but links RYBP to microRNA mediated regulatory processes too. DDX5 is important for processing several microRNA, including *miR-125b*. The 3’UTR of *Rybp* harbors a *miR-125b* binding site, and the transcription of the *Rybp* gene is repressed by *miR-125b*. Taken together, it seems that *Ddx5* loss of function mutation promotes reprogramming by upregulation of *Rybp* through the suppression of *miR-125b* processing [71]. The *miR-125b*-RYBP-DDX5 axis might be more important than anticipated because *miR-125b* is also involved in the fast and effective process of somatic cell reprograming, when somatic cell nuclei are transferred into an oocyte [235]. *miR-125b* is strongly expressed in early mammalian embryos, and a high level of *miR-125b* is found in oocytes, able to effectively repress *Rybp* expression, which might facilitate somatic cell reprogramming.

## 8. Methods

### 8.1. ChIP-seq Analysis

Published ChIP-seq raw data from RYBP-GSM1041375, RING1B-GSM1041372, PCGF6-GSE84905, and RAR–GSM482749 were downloaded from Geo Database (https://www.ncbi.nlm.nih.gov/geo/) and analyzed for their corresponding gene targets.

### 8.2. RNA-seq Analysis

RNA-seq raw data were either downloaded from Geo Database (https://www.ncbi.nlm.nih.gov/geo/) for RYBP-GSE83134 [94] or downloaded the analyzed upregulated and downregulated list from published data for *Yaf2* and *Rybp* from Zhao et al., 2018 [125]. Raw data from GSE83134 was analyzed based on log_2_ fold change values and those genes with at least 1.5 times upregulation were used for further analysis. Venn diagrams were made using Venn tool from Bioinformatics and Evolutionary Genomics webpage (http://bioinformatics.psb.ugent.be/webtools/Venn/). Gene ontology (GO) enrichment analysis was performed by uploading the gene list into GO enrichment analysis tool (http://geneontology.org) and the corresponding output was used to plot a bar graph using the predicted complementing GO terms in the Y-axis and the enriched number of genes in the X-axis. Scatter plot mapping the upregulated and downregulated hits from our previously published data [126] was used using raw counts of hits corresponding to a gene from both *wild type* and *Rybp^-/-^* ES cells. The genes with no difference in their reads comparing the wild type to *Rybp^-/-^* can be seen close to the trend-line.

## 9. Closing Remarks

RYBP is a major contributor to the regulation of germ-cell-specific gene expression. It can repress germ cell genes as a member of ncPRC1.6, or activate them directly, by cooperating with other transcription factors like E2Fs and OCT4, or indirectly, upregulating *Nanog*, *Kdm2b*, or influencing the expression of different RA pathway members. RYBP alters the ubiquitin-mediated degradations of many proteins and stabilizes important regulators of germ cell fate, like FANK1 and RING1. Changing stability and availability of the RYBP interactors and targets all lead to alterations in germ-cell-specific apoptotic processes, meiosis induction and timing of germ cell differentiation. *Rybp* is connected to somatic and nuclear transfer mediated reprogramming processes through its interaction with *Ddx5* and *miR-125b*. Moreover, as factor altering the efficiency of double-strand break repair, RYBP can also play a direct role in regulating meiotic processes and checkpoints. Taken together, Rybp can be described as a new recruit to the orchestra of germ cell regulators. 

Further biochemical and genetic studies need to answer important questions, which still remained unanswered. Rybp is a dosage-dependent regulator [236]. The differences in the strength of bindings with different partners and the availability of the RYBP protein can integrate important events of differentiation like meiotic entry, double-strand DNA repair, and apoptosis. Developing novel germ-cell-specific conditional *Rybp* mutants in mouse and analyze their phenotypes in testes and ovaries will give a better insight into the in vivo role of Rybp in this process. Still, the in vitro stem-cell-based model systems will remain equally powerful tools in order to specify further the molecular function of RYBP in germ cell development.

## Figures and Tables

**Figure 1 genes-10-00941-f001:**
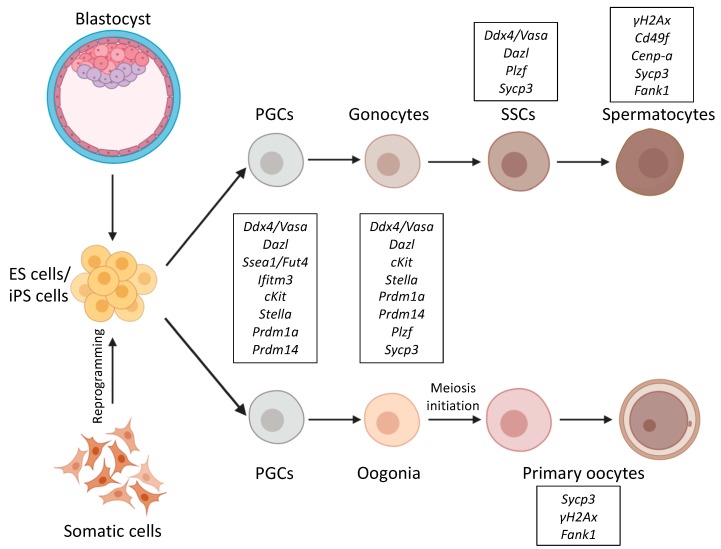
Schematic representation of differentiation pathways from ES/iPS cells into germ cells. ES cells are derived from the mouse blastocyst, and iPS cells are reprogramed from adult somatic cells. Both ES/iPS cells can be differentiated into PGCs and further towards either spermatocytes or primary oocytes. Germ cell markers for the corresponding stages are marked as framed. Certain germ-cell-specific genes, like the ones coding for germ granule components like *Ddx4/Vasa* are active all through germ cell development, while others like *Fank1* only expressed at distinctive phases of germ cell development. Abbreviations: ES: Embryonic stem, iPS: Induced pluripotent stem, PGCs: Primordial germ cells, SSCs: Spermatogonial stem cells.

**Figure 2 genes-10-00941-f002:**
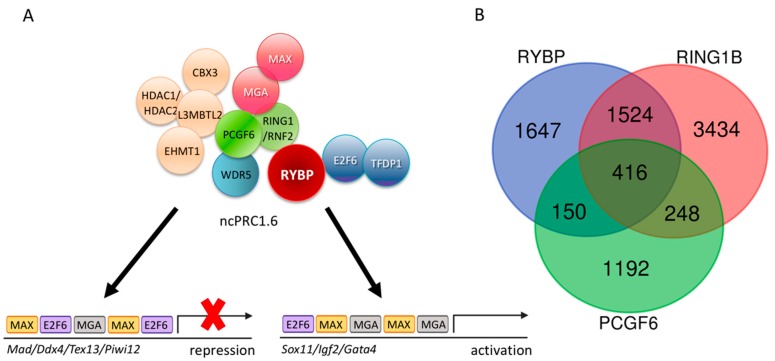
The non-canonical Polycomb Repressor Complex1.6 (ncPRC1.6) complex is targeted by DNA binding subunits and its core members co-exist only on a subset of targets. (**A**) Schematic representation of the ncPRC1.6 complex showing both activation [105] and repression activities [106]. (**B**) Venn diagram showing the common targets of Ring1 and YY1 binding protein (RYBP), Ring finger protein2 (RNF2/RING1B), and Polycomb group ring finger 6 (PCGF6). Chromatin Immune Precipitation followed by whole genome sequencing (ChIP-seq) raw data derived from Geo Database: ID GSM1041375 for RYBP, GSM1041372 for RING1B, and GSE84905 for PCGF6. List of the 65 common upregulated genes is found in Appendix A.

**Figure 3 genes-10-00941-f003:**
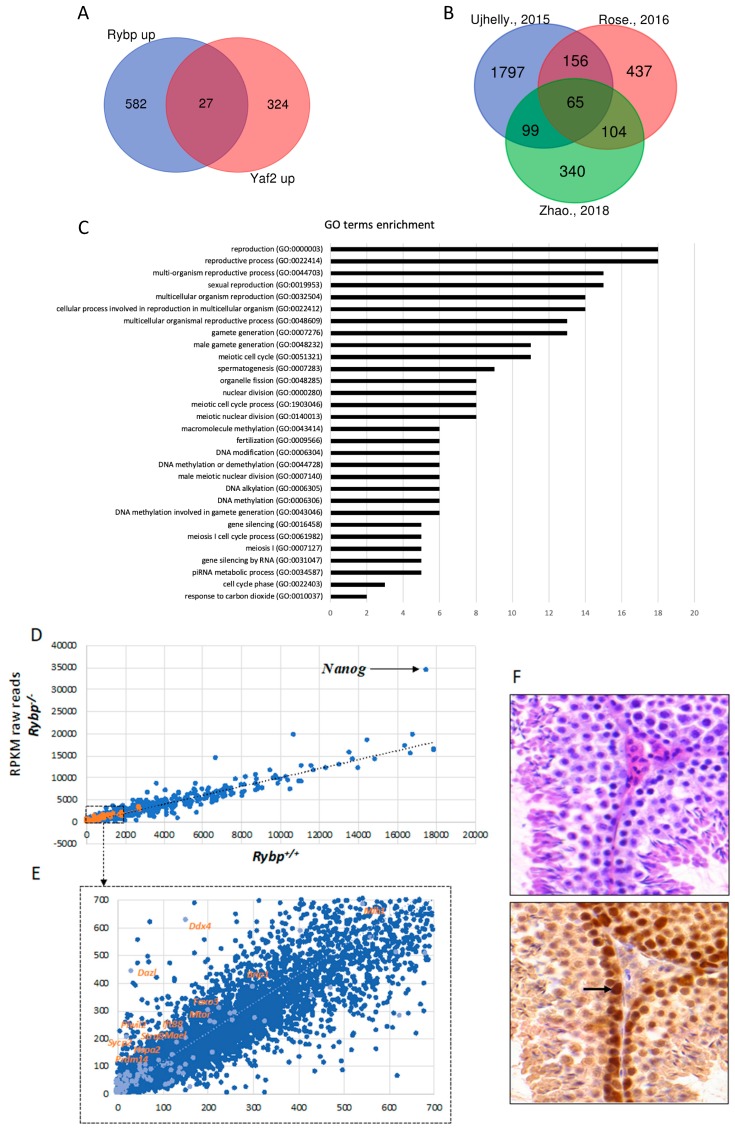
RYBP regulates germ-cell-specific genes and is expressed in the adult mouse testis. (**A**) Common upregulated targets of Ring1 and YY1 binding protein (RYBP) and YY1 associated factor2 (YAF2). List of genes upregulated in the *Rybp^-/-^* and *Yaf2^-/-^* backgrounds, according to Zhao et al., 2018 [125] was used to produce Venn diagram showing the 27 common *Rybp* and *Yaf2* upregulated genes. (**B**) Venn diagram showing the 65 targets of RYBP revealing top hits connected to germ cell development and meiosis. List of significantly upregulated genes in the *Rybp* mutant background from independent experiments, according to Ujhelly et al., 2015 [126], Rose et al., 2016 [94] and Zhao et al., 2018 [125]. (**C**) Gene onthology (GO) enrichment of the upregulated 65 targets of RYBP. GO enrichment analysis was performed for the 65 commonly upregulated genes in the *Rybp* mutant ES cells revealing top hits connecting to germ cell development and meiosis. (**D**) Key germ cell and meiotic genes are upregulated in the *Rybp^-/-^* background. Raw reads from genome-wide transcriptome analyses by Ujhelly et al., 2015 [126] was used to produce a scatter plot, plotting wild type *(Rybp^+/+^*) in the X-axis and *Rybp^-/-^* in the Y-axis. Highlighted dots in orange represent the germ-cell-specific genes upregulated in the *Rybp^-/-^* mutants. *Nanog*, which is highly upregulated in the *Rybp^-/-^* cells is indicated with a black arrow (Reads Per Kilobase Million, (RPKM): *Rybp^+/+^*-17524.93; *Rybp^-/-^*-34448: Fold change-1.96) (**E**) Highlighted region from Figure 3D (Dotted box) is enlarged to show germ-cell-specific genes. (**F**) RYBP is present strongly at the mouse adult spermatogonia. Immunohistochemical analysis of adult mouse testis stained with hematoxylin and eosin (purple) and anti-RYBP antibody (brown) (Merck Millipore, Cat.No: AB3637, Darmstadt, Germany) showing the presence of RYBP in the primary and secondary spermatozoa. Strong expression in the spermatogonial stem cells is indicated by a black arrow.

**Figure 4 genes-10-00941-f004:**
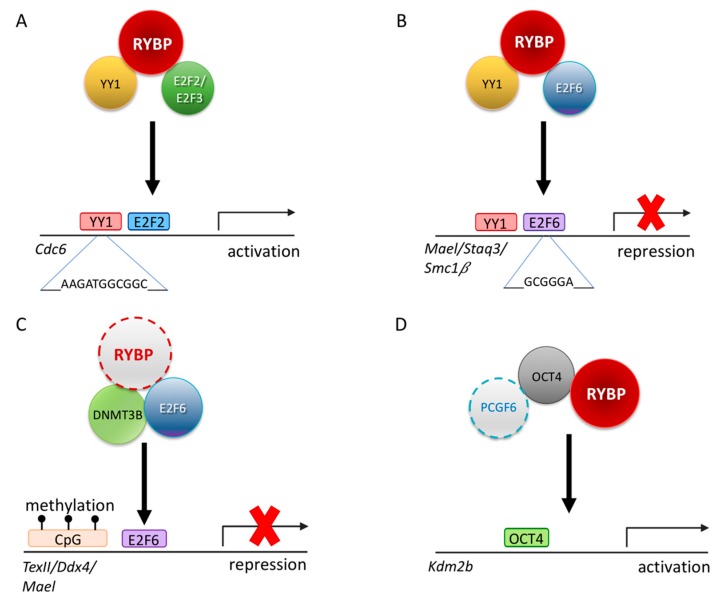
Possible regulatory mechanisms of RYBP driving germ cell development. (**A**) Schematic representation illustrating that RYBP can interact with E2F transcription factor 2 and 3 (E2F2/E2F3) and YY1 transcription factor (YY1) to transcriptionally regulate *Cell division cycle 6* (*Cdc6*) [89]. (**B**) Schematic representation illustrating that RYBP can interact with E2F6 and YY1 to repress genes such as *MaeI*, *Staq3* and *Smc1**β* [90]. (**C**) Schematic representation of a possible repression mechanism of RYBP via the E2F6 and DNA methyl transferase 3b (DNMT3B) on *Tex11*, *Ddx4* and *MaeI* locus by recognizing the methylation mark at the CpG islands [201]. (**D**) Schematic representation of PCGF6, RYBP and OCT4/POU5F1 acting synergistically in order to activate *Kdm2b* [71].

**Table 1 genes-10-00941-t001:** Comparative analysis of Chromatin Immune Precipitation followed by whole genome sequencing (ChIP-seq) binding sites of Ring1 and YY1 binding RYBP, RNF2/RING1B, PCGF6, and RAR on retinoic acid (RA) pathway members, upregulated in the *Rybp^-/-^* ES cells.

Gene Name	Transcriptome (*Rybp^+/+^*/*Rybp^-/-^*) Raw Counts	Fold Change	RYBP ChIP-seq (GSM1041375)	RING1B ChIP-seq (GSM1041372)	PCGF6 ChIP-seq (GSE84905)	RAR ChIP-seq (GSM482749)
*Adh1*	0/0.7	Inf	-	-	-	-
*Adh4*	0/5.65	Inf	-	-	-	+
*Rarb*	0/2.12	Inf	-	+	-	+
*Rxrg*	0/1.41	Inf	-	+	+	+
*Crabp1*	36.8/218.5	5.94	+	+	-	-
*Rdh14*	4.2/13.4	3.166	-	-	-	-
*Rxra*	29.7/38.2	1.285	-	+	-	+

Raw reads from genome-wide transcriptomics from wild type (*Rybp^+/+^*) and *Rybp^-/-^* embryonic stem (ES) cells reported by Ujhelly et al., 2015 [126]. ChIP-seq data for the binding targets of RYBP, RNF2/RING1B, PCGF6, and RAR (see Methods) were analyzed for upregulated RA pathway members in the *Rybp^-/-^* ES cells. ‘+’ represents identified ChIP binding targets and ‘–‘ represents no binding. RYBP, RING1 and YY1 binding protein. ES, embryonic stem. PCGF6, Polycomb group ring finger 6.

**Table 2 genes-10-00941-t002:** Comparative analysis of Chromatin Immune Precipitation followed by whole genome sequencing (ChIP-seq) binding sites of RYBP, RNF2/RING1B, PCGF6, and RAR on meiotic genes upregulated in the *Rybp^-/-^* ES cells.

Gene Name	Transcriptome (*Rybp^+/+^*/*Rybp^-/-^*) Raw Counts	Fold Change	RYBP ChIP-seq (GSM1041375)	RING1B ChIP-seq (GSM1041372)	PCGF6 ChIP-seq (GSE84905)	RAR ChIP-seq (GSM482749)
*Xlr4a*	0/42.42	Inf	-	-	-	-
*Xlr4b*	0/42.43	Inf	-	-	-	-
*Tex11*	5.65/223.44	39.50	+	+	+	-
*Dazl*	31.11/437.69	14.068	+	+	+	-
*Tex15*	31.11/280.01	9.00	+	+	-	+
*Xlr3a*	7.07/63.63	9.00	-	-	-	-
*Piwil2*	22.62/201.52	8.90	+	+	+	+
*Xlr3c*	1.41/12.02	8.50	-	-	-	-
*Tdrkh*	29.69/250.31	8.42	+	+	+	-
*Hormad2*	16.97/110.30	6.50	+	-	-	+
*Sycp2*	21.21/137.17	6.46	+	+	+	+
*Mov10l1*	55.15/335.87	6.090	+	+	+	+
*Piwil4*	5.65/33.94	6.00	-	-	-	+
*Mael*	87.68/415.77	4.74	+	+	+	+
*Ddx4*	151.32/623.66	4.120	+	+	+	+
*Tdrd1*	18.38/65.76	3.57	+	+	+	+
*Smc1b*	100.40/337.28	3.35	+	+	+	+
*Xlr3b*	9.89/32.52	3.285	-	-	-	-
*Cpeb1*	11.31/36.06	3.187	+	+	+	+
*Stra8*	48.08/152.73	3.176	-	-	-	+
*Boll*	7.07/20.5	2.90	+	+	+	+
*Meioc*	49.49/142.12	2.87	+	+	+	+
*Asz1*	2.82/7.77	2.75	+	+	-	+
*Sycp1*	251.73/412.95	1.640	+	+	+	-
*Prdm14*	52.34/84.14	1.60	-	+	-	+
*Sycp3*	322.44/381.83	1.184	+	+	-	+

Raw reads from genome-wide transcriptomics from wild type (*Rybp^+/+^*) and *Rybp^-/-^* ES cells reported by Ujhelly et al., 2015 [126] and ChIP-seq data for the binding targets of RYBP, RNF2/RING1B, PCGF6 and RAR (see Methods) were analyzed for upregulated meiotic targets in the *Rybp^-/-^* ES cells. Targets. which were identified in more than one independent experiment are included in the table. ‘+’ represents identified ChIP binding targets and ‘–‘ represents no binding.

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
