# Peer review of "Evolving Role of RING1 and YY1 Binding Protein in the Regulation of Germ-Cell-Specific Transcription"

_genes, 2019, doi:10.3390/genes10110941_

Round 1

Reviewer 1 Report

The review by Bajusz, et al deals with a regulatory protein, RYBP, involved in the non-canonical polycomb repressive complex and its role in regulating germ cell-specifying gene transcription. The authors attempt to put this regulation in the larger context of overall differentiation genes in germline stem cells and embryonic stem cells, as a contrast to somatic cells. The review attempts to be a comprehensive treatise for early gene events for germ cell specification, but it centers heavily on the repressive polycomb complex regulation. There is a laudible effort to describe the functions and molecular understanding of all the transcription factors, histone modifiers, E3 ubiquitin ligases, and coordinating modulator proteins (like RYBP) involved in each pathway. Unfortunately, the text often rambles in progression through gene regulatory sets with little topical focus except, perhaps, historical discovery. Simply stated, there are long blocks of explanatory text and lots of molecular detail (e.g. Section 3, lines 104-165), but it is left to the reader to decide the logical direction being taken. It would be considerably more helpful to periodically summarize the detailed descriptions and put them in an overall scheme for germ cell differentiation. Clearly depicted diagrams as Figures would be very helpful.

The review mentions post-transcriptional and mRNA/RNP granule regulation in passing, but then largely ignores its importance in germ cell biology. This is a rapidly developing area of germ cell specification/differentiation that has ironically long been known to have great importance for these cells in particular. Statements regarding formed mRNPs in embryonic germ cells and their localization seem to contradict one another, and there is no attempt to integrate mRNA regulation with the more central theme of transcriptional gene regulation. This seems to be an opportunity missed.

Individual Comments:

(Lines 76-84) The authors put forward “epigenesis” as the lone germ cell determinant in mammals and subsequent state that “no clear evidence is found that the localization of these determinants (mRNPs, nuage) is important…”. These statements appear to be contradicted by later descriptions (lines 104-113) of the critical maternal determinants (“RNAs and proteins”) including Vasa, Nanos, Piwi, for which localization is in some cases critical, as well as the “cytologically distinct assembly of electron-dense RNA-protein granules and mitochondria”. The authors should do a better job of integrating these two concepts into a single one. The author’s use of experimental transcriptome data for Rybp- ES cells from 3 independent studies to make a case for polycomb regulation of germ cell specifying genes is not convincing. Whereas each study found between 600-2000 upregulated genes in Rybp’s absence, only 65 genes were in common. That means that more than 95% of the genes discovered in each study were likely responding to ES cell growth conditions and environment rather than Rybp. This casts some doubt on the significance of the 65 common genes, regardless of their GO term enrichment in germ cell biology functions (also a very broad category). post-transcriptional mRNA regulation for germ cell differentiation is largely ignored. However, there is good evidence that key genes in the pathway (e.g. c-Kit), signaling molecules (RA) and pathways (Akt/mTOR) take part in critical mRNA translational control (See ref’s 1-3). In section 7.2 (lines 482-496) the authors describe roles for Rybp and FANK1 in functions associated with apoptosis, in a manner that appears unexpected. However, there are many known conventional uses for apoptotic proteins (e.g. caspases, Apaf1) and cell mechanisms that are common to germ cells and dying cells (see refs 4-6). There are many grammatical errors and word misusage in the manuscript that must be addressed. Examples: (Abstract, line 22) “propose a concept how”, instead “propose a model for how”; (lines 36-37) “..in tumors [4] it is not”, instead, new sentence after ref; (line 38) “modell”, instead “model”; (line 43) should be “Germ cells differ profoundly…”; (line 68) “preformistic” is not an English word, better “deterministic”; (line 54) “loose”; instead “lose”; (lines 590, 737) “double strand brake” should be “double strand break”, etc.

References:                                    

Hermann, B. P., Mutoji, K. N., Velte, E. K., Ko, D., Oatley, J. M., Geyer, C. B. and McCarrey, J. R. (2015). Transcriptional and translational heterogeneity among neonatal mouse spermatogonia. Biol Reprod 92, 54.

Serra, N., Velte, E. K., Niedenberger, B. A., Kirsanov, O. and Geyer, C. B. (2019). The mTORC1 component RPTOR is required for maintenance of the foundational spermatogonial stem cell pool in mice. BiolReprod 100, 429-439.

Galluzzi, L., Joza, N., Tasdemir, E., Maiuri, M. C., Hengartner, M., Abrams, J. M., Tavernarakis, N., Penninger, J., Madeo, F. and Kroemer, G. (2008). No death without life: vital functions of apoptotic effectors. Cell Death Differ. 15, 1113-1123. 

Author Response

We would like to thank the Editor and the Reviewer for the comments, critical thoughts and for the recommended articles regarding to our review article, entitled “Evolving role of RING1 and YY1 binding protein in the regulation of germ cell specific transcription”. We are grateful for the suggestions of the Reviewer and we hope that we managed to integrate all remarks in the report to improve the manuscript. Specific details of the changes that were made are detailed as follows:

We corrected all of the grammatical errors and misused words listed in the review report and rephrased some other sentences of the manuscript for better understanding.

In order to clarify the overall scheme of germ cell differentiation we extended the introductory part of mouse germ cell development (line113-136), and added a new figure (Fig1.) to show the different stages, in connection with the expressed marker genes. We focused more to the role of germ granules (line99-101, line118-134) in different developmental stages, and extended this introductory chapter, as it was advised.

We made new subheadings in Section 3 (line149, line159, line170, line 192, line251), where we are discussing the germ cell differentiation factors. We hope that this new structure makes this long and complicated section more understandable.

We tried to specifically focus our review to the function of RYBP in germ cell specific gene expression regulation. As RYBP is a known member of the Polycomb group of regulators, we thought that it might be useful to summarize the general role of Polycomb repressor complexes, as their role in germ cell specific gene regulation has been only revealed recently. The new findings discussed in this chapter partially contradicts to the classical histone code oriented chromatin structure compaction based models of Polycomb complexes. In the case of non-canonical Polycomb repressor complex1.6 (PRC1.6), which seems to play the most prominent role in regulating germ cell specific genes, the targeting is clearly sequence specific, and largely independent of the presence of repressive histone methylation mark. It was unexpected, and contradicted to the classical hierarchical recruitment model of PRC dependent gene repression. PRC1.6 dependent repression of germ cell specific and meiotic genes in somatic cells is important on its own, but at certain loci these complexes seemingly play a role in activation processes too. We wanted to emphasize the importance of these new findings in our work.

We thank for the reviewer for its comments and the recommended article about the connection of apoptotic proteins and their alternative role in independent germ cell specific or stress related functions. We included these arguments into the chapter about the apoptotic functions of RYBP in connection to FANK1 (line581-587).

As RYBP is an unstructured, “moonlighting” protein, it is capable to interact with many partners and can be involved in many ways regulating germ cell development from gene repression to apoptotic processes, from modulating ubiquitin mediated degradation of germ cell specific regulatory proteins to the micro RNA mediated regulatory processes. We wanted to emphasize that a protein with a well described function in parallel can play an independent role in many alternative process, interacting with different partners. While the differences in the strength of bindings with different partners and the availability of the protein alone can integrate the important events of differentiation like meiotic entry or apoptosis.

We also extended the part of the manuscript concerning the upregulated gene set in different Rybp mutant ES cells (Section 5 line390-398). We described the different origin of the ES cells and the types of mutations. One of the experiments were performed with conditional knockout cell line, we indicated that in the text too. We set a relatively high threshold for upregulation (2 times) many published datasets use lower values (1.3 times upregulation can also be acceptable in some cases). Many germ cell specific genes are fully repressed in the soma, accordingly their transcript level (Read Per Kilobase Millon, RPKM) is 0 in the dataset. In these cases if only a very few mRNA sequences are identified in the mutant cell line, the fold change become highly significant, but unfortunately in this category of mRNAs many neutral targets appear as well, which only increase the “noise” of the data. This is the reason why we thought that using the whole genome transcriptome data from independent experiments can resolve this issue. The GO term enrichment analysis program we used is specifically designed for identification of the enrichment of certain sets of genes in a sample gene set. Only 344 genes are included in the GO term of “Germ cell development” of the over twenty thousand mouse gene annotated. 6 of them are found in the set of 65 commonly upregulated genes.  

We think that the assumption that more than 95% of the genes discovered in each study were likely responding to ES cell growth conditions and environment rather than Rybp is possible but not likely. The fact that besides the germ cell genes there are also genes from other GO categories amongst the common hits, rather underlines the importance of Rybp in other processes as well then weakens its role in germ cell development. These GO categories include cell growth, as the reviewer pinpointed, but also include genes important is certain signaling pathways or organ development such as Tbx3, the gene important in anterior-posterior axis formation, limb, tooth and heart development. Moreover, Tbx3 mutations also occur in ulnar-mammary syndrome. We think that Rybp might be important for multiple processes due to its moonlighting functions and this does not necessary weakens its possible function in germ cell development. Finally, since wild-type cells were used as controls in all 3 transcriptome analyses, these genes are likely that affected by the lack of Rybp and not the given growth conditions.

We thank again to reviwer1 to the helpful suggestions, which promoted us to improve our manuscript. We hope that this form of the manuscript will be more satisfying.

Sincerely,

Dr. Izabella Bajusz

Biological Research Centre,

Temesvári krt. 62, Szeged 6726, Hungary

bajusz.izabella@brc.hu

Reviewer 2 Report

Manuscript describe role of (RYBP) RING1 and YY1 binding protein in regulation of transcription in germ line. Authors analyzed and compared published RNA seq data from Zhao et al. 2018 and other public databases.  Authors propose importance of RYBP in multilevel regulation network. They present 3 figure and 2 tables with list of genes and 218 references. Introduction mention fundamental germ cell differentiation factors (Pou5f1, Klf4, Nanog, Bmp4) in different model organism (C.elegans, Drosophila, mice) and relevant references. Article is written in an appropriate way with sufficient details. Authors deeply characterize ncPRC1.6 complex. RYBP can be substituted with homolog protein YAF2 in ncPRC1.6 complex or RYBP can independently bind OCT4. They compare similarities between three independent studies (Ujhelly et al. 2015 ; Rose et al. 2016 , Zhao et al. 2018) and identified 65 influenced genes.

I recommend this article to accept after minor revision.  Current state of the manuscript need improvements and I have some comments about this.

comments:

Will be beneficial to provide tables of targets which are influenced by key molecules described in the manuscript.

I recommend enlarge Fig. 2C and Fig 2E for better visibility.

Can author speculate about role of Fank1 in maternal germ cells? Because authors didn’t mention expression in oocyte which were published by Hwang et al. 2005

Author Response

We would like to thank the Editor and the Reviewer for the feedback on our review article, entitled “Evolving role of RING1 and YY1 binding protein in the regulation of germ cell specific transcription”. We are grateful for the suggestions of the Reviewer that have helped us to make the revised manuscript an improved version. Specific details of the changes that were made are detailed as follows:

We corrected the grammatical errors and misused words listed in the review report and rephrased a few sentences of the manuscript for better understanding.

Figs 2C and 2E have been enlarged for better visibility (now in Fig3C and Fig3E).

We thank for the recommended article by the reviewer. We included a new paragraph about the possible role of FANK1 in oocytes based on the article recommended by the reviewer (line568-580).

The reviewer advised us to include additional tables of the important targets mentioned in the review. In order to fulfill this requirement, we included two additional Supplementary datasets about the list of commonly upregulated 65 target genes and the detailed list of genes upregulated, which are connected to “Germ cell”, “Meiotic” functions or “Spermatogenesis” by GO terms (See Supplementary data1 and Supplementary data2).

Sustained expression of Nanog alone can induced germ cell fate, but the exact mechanism has not been analyzed transcriptomically in mouse yet. Three key targets of NANOG regulation described in Murakami et al. 2016., in the article, where this phenomenon of Nanog was originally described. Amongst these key NANOG targets Prdm14 (1.6 times), Dppa3 (1.91 times), and Dazl (14 times) upregulated in The Rybp-/- ES cells according to the dataset of Ujhelly et al. 2015. The threshold we set for “upregulation” is at least 2 fold increase in mutant background. Two out of these three targets are under this threshold, thus they did not come up as “upregulated target genes” in our list (See Supplementary data1). Unfortunately, genome-wide NANOG targets were only determined in human ES cells. We are attaching a new dataset (Supplementary data3), based on the overlapping targets of Rybp in mouse ES cells (based on the transcriptomics data) and the published Nanog ChIP binding sites found in human ES cells (Boyer at al 2005.). There are 1244 identified ChIP binding sites of NANOG identified by Boyer et al. We found no overlap between the 65 common targets of more than 2x upregulated in all three transcriptomes of Rybp mutant ES cells, but 81 genes are common between the 2117 upregulated genes published in Ujhelly et al 2015 and the 1244 NANOG bound target genes identified in human ES cells.

Based on the criticism and advices of the other reviewer we extended the introductory part of the review with more detailed description of germ cell differentiation and emphasized more the role of germ granules as well (Line99-134) and added a new Figure1.

We thank for Reviewer 2 again for the comments and suggestions that have led to the current, improved version of our manuscript.

Sincerely,

Dr. Izabella Bajusz

Biological Research Centre,

Temesvári krt. 62, Szeged 6726, Hungary

bajusz.izabella@brc.hu

REFERENCES

Murakami, K.; Günesdogan, U.; Zylicz, J. J.; Tang, W. W. C.; Sengupta, R.; Kobayashi, T.; Kim, S.; Butler, R.; Dietmann, S.; Surani, M. A. NANOG alone induces germ cells in primed epiblast in vitro by activation of enhancers. Nature 2016, 529, 403–407, doi:10.1038/nature16480 Ujhelly, O.; Szabo, V.; Kovacs, G.; Vajda, F.; Mallok, S.; Prorok, J.; Acsai, K.; Hegedus, Z.; Krebs, S.; Dinnyes, A.; Pirity, M. K. Lack of Rybp in Mouse Embryonic Stem Cells Impairs Cardiac Differentiation. Stem Cells Dev. 2015, 24, 2193–2205, doi:10.1089/scd.2014.0569. Boyer LA1, Lee TI, Cole MF, Johnstone SE, Levine SS, Zucker JP, Guenther MG, Kumar RM, Murray HL, Jenner RG, Gifford DK, Melton DA, Jaenisch R, Young RA. Core transcriptional regulatory circuitry in human embryonic stem cells.

Round 2

Reviewer 1 Report

The review article by Bajusz, et al now reads more clearly. The new section titles and organization into topics makes the content easier to follow. The new Fig. 1 is helpful for non-specialists. In addition, there were substantial improvements to the "2. Differentiation.." section. This text deals more clearly and openly with some issues surrounding PGC determination, particularly concepts of epigenesis and germ granules and their contributions.

There is now a very nice, short synopsis of germ granules; what they are and the roles they may play. One note, in line 231 the sentence should be corrected to say, "RNAs can be stored in the granules for weeks without being translated,..." (rather than "transcribed").

The added paragraph that states, in part, "Heterogeneity in the transcription pattern might be a fundamental feature of the germ line..." is a welcomed addition, and better explains the author's data set.

Finally, the mention of non-apoptotic roles for death-related proteins in germ cells seems to add an interesting twist to the review. It was nicely incorporated.

The revised version of this review is much improved.